# Latent classes of substance use and delinquency in a Swedish national sample of adolescents and associated risk factors

Clas Björklund[1]*, Fredrik Sivertsson[2], Jonas Landberg[1], Jonas Raninen[3,4], Peter Larm[1]

1 Department of Public Health Sciences, Stockholm University, Stockholm, Sweden, 2 Department of Criminology, Stockholm University, Stockholm, Sweden, 3 Department of Clinical Neuroscience, Karolinska Institutet, Stockholm, Sweden, 4 Center for Alcohol Policy Research, La Trobe University, Melbourne, Australia

* clas.bjorklund@su.se

## Abstract

### Background

Identifying underlying subgroups might be a way to examine the development of co-occurrent substance use and delinquency. The aim of this study was to identify latent classes of substance misuse and delinquency in adolescence and which general risk factors are associated with these classes.

### Methods

Data of two waves from a national representative Swedish birth cohort was used that consisted of 4,013 randomly selected adolescents (Male = 1,798, Female = 2,201, Missing = 14) Latent class analysis was used to identify classes of substance misuse and delinquency at age 17/18 and. logistic regression analysis was used to assess risk factors at age 15/16.

### Results

Identified classes were: "Low/abstainers" (74.80%, n = 2858, Male = 1191 Female = 1656, Missing = 11) which acted as reference, "Alcohol only" (22.21%, n = 849, Male = 420, Female = 426, Missing = 3), "Polydrug use and crime" (2.15%, n = 82, Male = 52, Female = 30) and "High crime" (0.84%, n = 32, Male = 30, Females = 2). Factors associated with belonging to any classes engaging in substance use and delinquency were lower parental support, supervision, peer problems, and higher conduct problems, sensation-seeking behavior, distrust in society, and truancy.

which permits unrestricted use, distribution, and reproduction in any medium, provided the original author and source are credited.

**Data availability statement:** The data for this study are managed by Karolinska Institutet. The Futura01 survey include sensitive information on individuals, as defined by the Swedish Personal Data Act (Personuppgiftslagen, PUL, SFS 1998:204; § 13). Restrictions apply to the availability of these data, which were used under license for this study. Data are available from registrator@ki.se given approval from the Swedish Ethical Review Authority (registrator@etikprovning.se). Access to the data for the purpose of this study was based on membership of a research team with ethical approval for analyses of data from Futura01. Researchers who qualify by getting ethical approval from Swedish authorities can request access to data from the data holder. Data is stored according to the Archives Act (1990:782) and the regulations and general recommendations of the Swedish National Archives concerning the disposal of documents in research activities conducted by government agencies (RA-FS 1999:1).

**Funding:** The author(s) received no specific funding for this work.

**Competing interests:** The authors have no competing interests.

## Conclusions

Most people did not engage in substance use or delinquency. When accounting for less frequent behaviors such as normative adolescent drinking and one-time events of crime and drug use, about 3% of the population engaged in co-occurring substance use and delinquency. Several different factors from several domains where related to belonging to a class that used substances and/or engaged in delinquency. There were indications that the most extensive users and offender displayed a wide variety of severe level risk factors, which could have implications for targeted interventions. Though, statistical power was a problem and future research should use larger samples or alternative methods.

## Introduction

Substance use and delinquency constitute a burden for public health, since substance use is associated with premature death, infectious diseases and poorer mental health [1, 2], while being a victim of crime is associated with premature death and negative health conditions [3, 4]. Individuals who are involved with both drug use and criminal behavior are also at a risk of developing health issues related to a long-term antisocial lifestyle such as hepatitis, pancreatitis, and sexually transmitted diseases [5]. Both behaviors are common. It is estimated that there are 1320.8 alcohol use disorders per 100 000 people and between 52.1 to 289.7 various drug use disorders per 100 000 people globally [1]. Also, interpersonal violence is the fifth leading cause globally for injuries [6]. It has also been reported that a fifth of the population in Sweden has been the victim of a crime [7].

Substance use and crime tend to co-occur in an individual. It is estimated that a quarter of incarcerated prisoners meet the criteria for alcohol use disorder and close to a third of prisoners meet the criteria for some type of drug use disorder [8]. Also, the prevalence of substance use disorders among individuals released on parole or on probationary sentencing are estimated to up to nine times higher than in individuals who have not had any contact with the criminal justice system [9].

Co-occurring substance use and delinquency might be more serious compared to only substance use and delinquency on their own. Co-occurrence of both behaviors is associated with worse mental and somatic health outcomes, as well as higher risk of recurrent substance misuse and reconviction, as compared to single behaviors [10]. Drug use can also contribute to an escalation of criminal activity and frequency of crime is higher during periods when use is also high. Substance use can also prolong the trajectory of criminal behaviors. Substance users seem to take a longer time to phase out of criminal activities than non-users [11–13]. Thus, it is important to study the different ways in which substance use and delinquency tend to develop into co-occurring behaviors.

The association between substance use and crime has been explained by pharmacological effects that substances have on executive functions, increasing the risk

of violence and other crime [14, 15]; that individuals commit crimes to finance addiction [15, 16]; that criminals engage in subcultures where substance use is promoted [12,17]; or that substance use and delinquency develop from similar risk factors [18].

From a developmental perspective, it seems that adolescence is a key period. This is the life-stage where both substance use and delinquency often have their onset, progress and peak through [19, 20]. It is also common that adolescents tend to engage in multiple types of risk behaviors and this might be considered a normative part of adolescence [21]. It is estimated that about 11% of Swedish 18-year-olds fulfil the criteria for substance use disorder and a considerable part are at risk of fulfilling criteria for such a disorder [22]. Also, it is estimated among Swedish adolescents that by age 19 almost a fifth of males and 5% of females have been registered at least once for a criminal act [20]. Thus, it seems that adolescence is a period where one is susceptible to the risk of developing substance use and delinquent behaviors. Common risk factors in adolescence are familial problems and negative parenting styles [23–25], antisocial peers, and peer pressure [17,18,23,25], mental health problems such as hyperactivity, conduct problems and having symptoms of conduct disorder [25–29] and problematic personality traits like impulsivity, sensation seeking and psychopathic personality traits [27–30]. Though there is knowledge on adolescent risk factors for substance use and delinquency, there is less knowledge about the patterns and mechanisms through which substance use and delinquency tend to combine into co-occurring behaviors in this stage of life. One approach to explore such processes is to find underlying subgroups of patterns of substance use and delinquency and which factors are associated with these patterns.

Many studies have identified latent classes of substance users. Tomczyk and colleagues [31] suggest in their literature review that common latent classes among adolescents include the largest class with low or no use of substances, a class using only one type of substance, usually alcohol, and a polysubstance class, using at least two psychoactive substances.

Latent classes of delinquent and criminal behaviors among adolescents indicate that most fall into a class of "low or no criminal behavior" and do not engage in delinquency to a high degree. Among those adolescents that do engage in delinquency a "low involvement in crime" class that generally refrains from violent crime, and a class of "high frequency and high diversity of crimes" has been identified [32–34].

For the development of co-occurring substance use and delinquency, classes of both substance use and delinquency are essential. However, only a few studies have taken this approach. One study on American violent offenders aged 12–25, found that almost 40% had indications of different types of substance use disorders, a larger group of alcohol and marijuana use disorders and a smaller for polysubstance use disorder [35]. Another study examined different classes of Swedes born between 1950–1993 registered for drug use by using the Swedish medical and crime registers. Two classes who engaged in criminal behavior, one larger with low frequency of drug use and one smaller with high frequency of drug use, and two classes who only engaged in drug use, one larger with lower frequency and one smaller with higher frequency, were identified [36]. Other studies have also identified classes of substance use and delinquency among risk populations such as inhalant users [37] or youths involved in diversion programs [17]. Since these studies are examining risk populations, it is not clear how these results translate to general population of adolescents. One study identified three latent classes among 548 adolescents aged 12–17 years recruited from two schools in Australia, where the most adapted class engaged in underage alcohol use only, whereas two other classes that were more heavily engaged in delinquency also used illicit drugs [38]. This study used dichotomized items (yes/no), neglecting frequency of behaviors. Consequently, knowledge of subgroups that engage in both substance use and delinquency is scarce, particularly as derived from large national representative samples.

Thus, the aim of the present study was to identify latent classes of substance use and delinquency in a national representative Swedish sample and to identify general developmental risk factors for the latent classes. Research questions included: (1) Which latent classes of substance use and delinquency can be identified in adolescence? (2) Which developmental risk factors are associated with each latent class in adolescence?

## Materials and methods

### Participants and procedure

Data from a longitudinal national birth cohort, known as Futura01, was used. In 2017 (T1), 500 Swedish schools were selected randomly, 343 of which agreed to participate (68.6%). Students in 9th grade (15–16 years) and born in 2001 were targeted, with one class per school randomly selected. Of 6,777 eligible students, 5,549 (81.9%) agreed to sign a written informed consent and to fill out a self-administered paper-and-pen questionnaire during school hours (between March and May). As reported elsewhere, the schools that participated did not differ regarding parental education, immigrant background or average grades compared to schools that did not participate [39]. Consent was only collected from the participants themselves and not from parents or legal guardians. According to the Swedish Act (2003:460) concerning the ethical review of research involving humans, individuals by the age of 15 are legally allowed to consent to participate in research without informing parents or legal guardians [40]. In 2019, when the participants were 17–18 years old, a web-based follow-up (T2) was administered (between March and June) and 4,013 (Male = 1,798, Female = 2,201, Missing information about sex = 14) responded (response rate of 72.3% of those that responded at baseline) and was included in the sample. Participants at T2 differed slightly from those at baseline, e.g., higher proportion of girls, higher sensation-seeking, aggressivity, and lesser alcohol consumption [41]. The project has been approved by the Ethical Review Board in Stockholm (Dnr 2017/103–31/5).

### Measurements

**Outcomes.** The outcomes of substance use and delinquency were from T2. All variables have been measured with self-report. Substance use consisted of *alcohol*, *cannabis*, and *other drugs*. Alcohol was measured with three items targeting consumption from the Alcohol Use Disorder Identification Test (AUDIT-C) which gives a score of 0–12, (α=0.76) [42]. Illicit drugs were measured via a battery of questions. The item used for different drugs was "How many times, during the latest 12 months, have you used any of the following?". Drugs that were included were 'marijuana', 'hashish', 'cocaine', 'spice/similar smoking substances', 'amphetamines', 'LSD/other hallucinogens', and 'other narcotics'. There was seven response alternatives which ranged from '0 times' (score = 0) to '40+ times' (score = 6). Illicit drugs were categorized into *cannabis* and *other drugs*. *Cannabis* contained marijuana and hashish (total score = 0–12, α=0.81, two items). *Other drugs* included the use of the other narcotics listed (total score: 0–30, α=0.77, five items). Delinquency was measured with questions targeting seven types of crime within the last twelve months, as developed by the Swedish National Council for Crime Prevention [43]. Four response alternatives varied from never (score = 0) to 5 + times (score = 3) for each type of crime. Delinquency was categorized into *violent crime* which included assault and illegally carrying a weapon (total score = 0–6, α=0.48), *theft* which included burglary, shoplifting and bike theft (total score = 0–9, α=0.48) and *property crime* which included vandalism and arson (total score = 0–6, α=0,56).

**Risk factors.** Risk factors were from T1 and categorized into the domains of (1) Family factors, (2) Mental health problems, (3) Personality traits, (4) Peer factors, and (5) Other factors. Family factors contained *parental support*, *parental supervision* and *parental rules* measured by two items each. *Parental support* targeted how often the participant received warmth and support from their parents (α=0.89). *Parental rules* targeted how often parents set rules for the participant (α=0.75). *Parental supervision* targeted whether the parents knew where the participant was and with whom (α=0.76). Each item had five response alternatives that ranged from 'almost always' (score = 0) to 'almost never' (score = 4). These items were reversed in coding (total score: 0–8, example item: "My parents know where I am during evenings"). 'Mental health problems' were measured using five-items subscales from the Strength and Difficulties Questionnaire (SDQ) [44], with three response alternatives for each item ranging from 'not true' (score = 0) to 'certainly true' (score = 2, total score: 0–10). These subscales included *emotional problems* (α=0.70, example item: I worry a lot), *conduct problems* (α=0.51, example item: I am often accused of lying and cheating), and *hyperactivity* (α=0.70, example item: I am easily distracted).

Personality traits included *impulsivity*, which was measured by five items from the Brief Barrat Impulsivity Scale (BIS-Brief) [45], with five response alternatives for each item ranging from 'strongly disagree' (score = 0) to 'strongly agree' (score = 4) (total score: 0–20, example item: "I easily get angry", α=0.73); *sensation seeking*, which was measured with six items from the Brief Sensation Seeking Scale (BSSS) [46], with five response alternatives for each item ranging from 'strongly disagree' (score = 0) to 'strongly agree' (score = 4) (total score: 0–24, example item: "I sometimes take risks because it is fun", α=0.88); and *callous-unemotional traits*, which were measured with three items from the Inventory of Callous-Unemotional Traits (ICU) [47], with three response alternatives for each item ranging from 'not at all true' (score = 0) to 'definitely true' (score = 3). Originally, this variable contained four items, but inter-item correlation showed one of the items to be uncorrelated with the others. This item was excluded, leaving three items. (total score: 0–9, α=0.52, example item: "I care about how others feel" (reverse item)). Since Cronbach's alpha was considered low for this factor, the mean inter-item correlation was also checked. This value was 0.27 and was within acceptable ranges [48]. 'Peer factors' included *peer problems*, which was measured by a five-item subscale of the SDQ (total score: 0–10, example item: "I am often by myself. I often do things alone", α=0.56) and being the perpetrator of bullying, which was measured by one item with a no/yes response. 'Other factors' included *distrust in society*, which consisted of five items targeting trust in various instances of society such as government and parliament, media, and researchers and experts, with four response alternatives varying from 'very high trust' (score = 0) to 'no trust' (score = 3) (total score: 0–15, α=0.76, example item: "How high is your trust in the judicial system?",). *Truancy* was measured by one item – whether the respondent had been truant from school – with six response alternatives varying from 'never' to 'several times per week'. *Truancy* was dichotomized into 'no/yes'.

## Analysis

Latent Class Analysis (LCA) was conducted in Stata 18 [49] to identify different subgroups of substance use and delinquency at T2. The z-scores for violent crime, property crime, theft, alcohol, cannabis and other drugs were entered as indicator variables. Missing values were deleted listwise. The highest missing value was for other drugs (4.23%, n = 170). The best model fit (number of classes) was determined by Akaike's information criterion (AIC) and Bayesian information criterion (BIC), where smaller values indicate better model fit [50]. In addition to these, log-likelihood was also examined where higher values indicate a better fit. Also, considering previous literature, smaller classes who engaged in substance use/delinquency were expected. Though, there was no official cutoff for how small classes could be. A correlation matrix (Supplemental material online) was created to check for correlations among indicators. Some have suggested that correlations greater than 0.5 among indicators might create statistical problems with multicollinearity that can lead to emergence of spurious classes [50].

To see what factors at T1 predicted classes at T2, pairwise logistic regression analysis was used. For selecting variables for the regression-model, ANOVA was used (chi-square for categorical variables) to identify significant differences between latent classes. These steps of the analysis were performed in IBM SPSS 28 [51] Logistic regression analyses, where factors with the highest p-value were removed stepwise until only significant factors remained, where then conducted. This was to remove insignificant predictors and reduce the number of predictors to minimize risk of potentially overfitting the model. Biological sex was controlled for in every step of the analyses.

## Results

Correlation was checked among the indicator variables before the analysis to check for multicollinearity. The highest was between *theft* and *property crime* (r = 0.60, p < 0.05) followed by *cannabis* and *other drugs* (r = 0.58, p < 0.05) which both exceeded 0.5. However, these indicators were deemed theoretically important and in line with the objective of the study and therefore included. The best latent class structure at T2 was identified based on model fit indices (AIC and BIC) which are shown in Table 1. A four-class solution best fitted the data.

**Table 1. Goodness of fit – measures of number of latent classes.**

| No. of classes | AIC | BIC | Log likelihood |
|---|---|---|---|
| 1 | 64637.05 | 64712.03 | -32306.53 |
| 2 | 57386.34 | 57505.06 | -28674.17 |
| 3 | 53764.12 | 53926.57 | -26865.06 |
| 4 | **52171.21** | **52377.40** | **-26052.60** |
| 5 | 52983.06 | 53226.74 | -26452.53 |

*Note: The chosen class solution is in bold.*

A description of the classes as marginal means of z-scores and also raw means is shown in Table 2. The largest class, which comprised 74.80% of the sample (n = 2858, Male = 1191, Female = 1656, Missing = 11), had the lowest scores on all outcomes and was named "Low/abstainers". This class was also selected as a reference group for the regression analyses as this was the class with the lowest amount of substance use and delinquency of all classes. The second largest class that emerged, which was 22.21% of the sample (n = 849, Male = 420 Female = 426, Missing = 3), was named "Alcohol only" with the highest alcohol use of all classes, but lower scores on everything else than other classes except for "Low/abstainers". The third largest class identified, which consisted of 2.15% (n = 82, Male = 52, Female = 30), was named "Polydrug use and crime" and scored in comparison to other classes high on *alcohol*, the highest on *cannabis* and *other drugs*, and the second highest on all crime types. The smallest identified class, which was 0.84% (n = 32, Male = 30, Female = 2), was named "High crime" which scored the highest on all crime types, the third highest on *alcohol* and the second highest on *cannabis* and *other drugs*. For visual representation see Fig 1.

Risk factor scores (means and percentages) at T1 for latent classes are shown in Table 3, with post-hoc differences from the Low/abstainers class identified by ANOVA and Chi-square tests. In general, the risk factor profiles of "Alcohol only", "Polydrug use and crime" and "High crime" differed from "Low/abstainers", with less favorable scores (i.e., lower scores on protective factors within "Family factors" and higher on risk factors within the other domains), except for *emotional problems* where only the "High crime" class had lower scores. *Parental rules* did not differ significantly from "Low/abstainers" among any of the groups and was excluded from the regression analysis for all classes.

Significant risk factors in Table 3 were included in pairwise multiple logistic regression analyses with "Low/abstainers" as the reference group, in order to identify which risk factors are uniquely associated with latent classes at T2. Non-significant factors were removed stepwise from the analysis until only significant risk factors remained for each class. The excluded factors after ANOVA tests and regression analyses were; *parental rules*, *emotional problems*, *impulsivity*,

**Table 2. Parameter estimates including marginal means of z-scores/raw means for latent classes.**

| Z-score/raw means | Low/abstainers | Alcohol only | Polydrug use and crime | High crime | Wald test |
|---|---|---|---|---|---|
| | 74.80%<br>(n = 2858) | 22.21%<br>(n = 849) | 2.15%<br>(n = 82) | 0.84%<br>(n = 32) | |
| Violent crime | -0.10/0.10 | 0.28/0.23 | 0.85/0.88 | 2.78/2.66 | 846.20 |
| Theft | -0.19/0.07 | 0.21/0.41 | 1.29/1.16 | 5.71/4.38 | 2070.64 |
| Property crime | -0.15/0.01 | 0.07/0.11 | 0.72/0.35 | 8.34/3.31 | 6016.53 |
| Alcohol | -0.50/0.82 | 1.28/4.72 | 1.20/4.35 | 1.00/3.94 | 5371.43 |
| Cannabis | -0.22/0.03 | 0.08/0.50 | 5.62/8.48 | 0.86/1.59 | 8693.89 |
| Other drugs | -0.12/0.01 | -0.05/0.08 | 3.69/3.71 | 2.35/2.41 | 1927.72 |

*Note. all Wald tests were significant at p < 0.05*

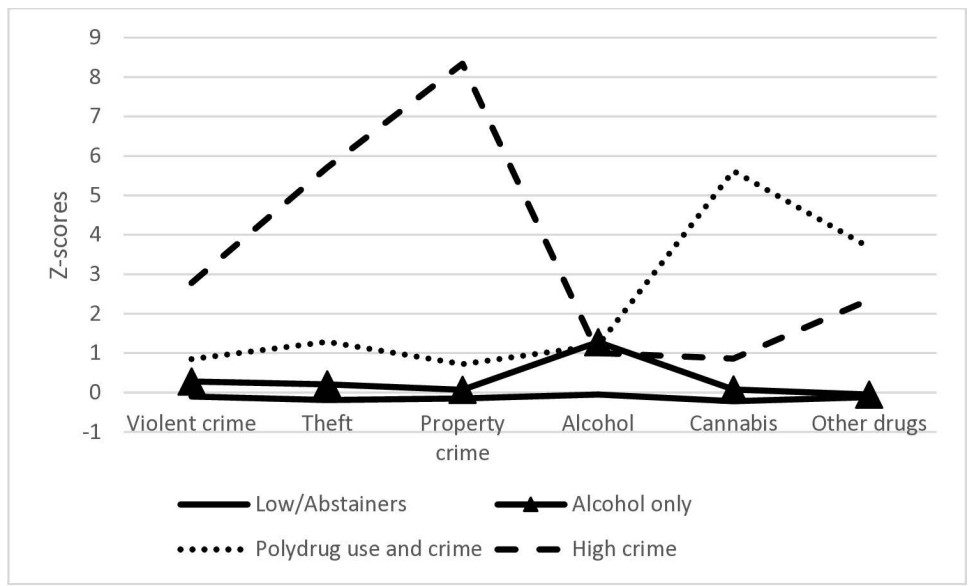

Note: Y-axis= z-scores of estimator variables. X-axis=estimator variables

**Fig 1. Line chart of z-scores of estimator variables grouped by latent classes.**

**Table 3. Risk factor profiles of latent classes presented as mean levels or percentages.**

| | Low/ abstainers (n = 2858) | Alcohol only (n = 849) | Polydrug use and crime (n = 82) | High crime (n = 32) | F value/Chi-square value |
|---|---|---|---|---|---|
| Family factors | | | | | |
| Parental rules | 4.12 | 4.20 | 3.77 | 3.92 | 0.86 |
| Parental supervision | 7.20 | **6.57** | **5.76** | **5.44** | 66.51 |
| Parental support | 6.99 | 6.85 | **5.84** | **5.68** | 16.13 |
| Mental health problems | | | | | |
| Hyperactivity | 3.59 | **4.40** | **4.90** | **4.77** | 32.90 |
| Conduct problems | 1.40 | **1.95** | **2.47** | **3.20** | 50.74 |
| Emotional problems | 3.50 | 3.44 | 3.51 | **2.31** | 2.77 |
| Personality traits | | | | | |
| Impulsivity | 7.25 | **8.94** | **10.65** | 8.90 | 50.60 |
| Callous-unemotional traits | 1.50 | **1.95** | **2.79** | **3.06** | 46.48 |
| Sensation seeking behavior | 9.04 | **12.75** | **16.68** | **13.64** | 127.64 |
| Peer factors | | | | | |
| Peer problems | 2.23 | **1.79** | 2.00 | 2.04 | 16.02 |
| Bullying others (% yes) | 2.50 | **6.1** | **15.70** | 6.70 | 51.22 |
| Other factors | | | | | |
| Truancy (% yes) | 20.6 | **33.0** | **61.7** | 31.3 | 118.56 |
| Distrust in society | 5.55 | **6.22** | **6.58** | **7.72** | 18.52 |

Note: Bold = significant differences (p < .05) from low/abstainers by ANOVA post hoc (means) and Chi-square test (percentage).

*callous unemotional traits* and *bullying others* for all classes; *parental supervision* for "High crime"; *parental support* for "Alcohol only"; *hyperactivity* for "Polydrug use and crime" and "High Crime; *conduct problems* for "Polydrug use and crime; *sensation seeking behavior* for "High crime"; *peer problems* for "Polydrug use and crime" and "High crime"; *truancy* for "Alcohol only" and "High crime"; and *distrust in society* for "Alcohol only" and "Polydrug use and crime". For belonging to the "Alcohol only" class, higher *parental supervision* and *peer problems* were associated with decreased probability, whereas higher *sensation seeking behavior*, *conduct problems* and *hyperactivity* were associated with increased probability. For belonging to the "Polydrug use and crime" class, higher *parental support* and *parental supervision* decreased the probability, whereas higher *sensation seeking behavior* and *truancy* increased the probability. For belonging to the "High crime" class, higher *parental support* decreased the probability, whereas higher *conduct problems* and *distrust in society* increased the probability. (see Table 4.).

## Discussion

The aim of the present study was to identify latent classes of substance use and delinquency in adolescence and to identify risk factors for each class. Four latent classes were identified.

Firstly, the largest class was "Low/abstainers" who participated in very little or almost no substance use and delinquency. This in line with previous research, as most people in the general population do not engage in any form of problematic substance use or criminal activity [31, 32]

Secondly, three classes engaging in substance use and/or delinquency were identified. Almost a quarter were classified into the "Alcohol only" class, where members displayed the highest alcohol consumption of all classes. This is also a commonly identified latent class [30]. Adolescence is the period where drinking, especially binge drinking, most often escalates and peaks. Among Swedish 18-year-olds it is estimated that almost a quarter has displayed either harmful use,

**Table 4. Associations between risk factors at T1 and latent classes at T2 calculated with multiple logistic regression analyses, providing Odds Ratios (OR) with 95% confidence intervals in brackets.**

|  | Alcohol Only (n = 849) | Polydrug use and crime (n = 82) | High crime (n = 32) |
|---|---|---|---|
| **Family factors** |  |  |  |
| Parental supervision | **0.88 (0.83-0.93)** | **0.87 (0.78-0.98)** | – |
| Parental support | – | **0.87 (0.78-0.96)** | **0.80 (0.68-0.94)** |
| **Mental health problems** |  |  |  |
| Hyperactivity | **1.05 (1.01-1.09)** | – | – |
| Conduct problems | **1.08 (1.01-1.15)** | – | **1.50 (1.23-1.82)** |
| **Personality traits** |  |  |  |
| Sensation seeking behavior | **1.08 (1.07- 1.10)** | **1.23 (1.16-1.29)** | – |
| **Peer factors** |  |  |  |
| Peer problems | **0.82 (0.77-0.87)** | – | – |
| **Other factors** |  |  |  |
| Truancy | – | **2.83 (1.71-4.68)** | – |
| Distrust in society | – | – | **1.15 (1.02-1.31)** |

*Note: Bold = sig. p < .05, Low/Abstainers at T2 acted as the reference group in all analyses. Biological sex was controlled for in all analyses. For each latent class, the risk factor with the highest p-value was removed stepwise from the analysis until only significant risk factors remained.*

alcohol dependence or sub-threshold for alcohol dependence [22]. It is likely that the "Alcohol only" class identified in this study consists of adolescents with similar drinking behavior. It still not clear, however, which of these adolescent's drinking is constrained to the adolescent years and whose is continued into adult years. This class also did not engage in any extensive delinquency despite having the highest alcohol consumption. This is notable since alcohol has been previously implied to be a contributing factor to criminal behavior, especially aggression and violence [14]. However, some research imply that alcohol use does not cause this behavior in itself, but rather directs it onto things such as provoking or inhibitory cues [52]. Also, it has been implied that the most frequent and persistent criminal offenders are driven by different familial and pathological factors early on in life, such as family problems, mental health issues and problematic personality [53]. It may be that there are other factors which are more driving in delinquency than only alcohol consumption. This could be possible, since judging by the mean scores of risk factors among the classes that engaged in substance use and/ or delinquency, the "Alcohol only" class seemed to be the most well-adjusted. Though, it should be noted that this is not something that has been tested in this study and therefore remains speculative.

Further, a "Polydrug use and crime" class was identified with co-occurrence of multiple substances, both alcohol, cannabis and other drugs, and acts of delinquency. This class had high alcohol consumption, though they were still consuming less than "Alcohol only". This class also had the highest illicit drug use compared to the other classes, particularly cannabis. This class was also more likely to engage in delinquency than both the "Low/Abstainers" class and the "Alcohol only" class. Though this difference was not as large as between the "High crime" class and the other classes. Although previous studies have identified a polysubstance use group that tends to be smaller in numbers than those that only engage in one type of substance [30], the present study extends these findings by demonstrating that this group of polysubstance using adolescents also commit crime, but to a smaller extent as their level of delinquent acts was not as high as the "High crime" class.

Moreover, a "High crime" class was identified who scored the highest on all crime types, considerably higher for every crime type compared to "Polydrug use and crime" who was the second highest for crime among the classes. This class also engaged in co-occurring behaviors, namely consuming alcohol, illicit drugs and committing crimes. The alcohol consumption of the "High crime" class was also considerably higher than for the "Low/abstainers" class. Also, this class had the second highest illicit drug use. In the general population, the number of individuals who display a pattern of high frequent delinquent behaviors is often smaller [32]. It has also been established that a small group of individuals commit a disproportionate number of crimes in the general population. An example of this phenomenon is that 68% of violent crimes in Sweden can be attributed to 1% of the population [54]. Another example of this is that of the life course persistent offender in Moffit's taxonomy. A group of offenders that is very small in size, but commit a disproportionately high frequency of criminal offences and also begin their offending in early adolescence and childhood [53]. This can correspond to the "High crime" class that was identified in this study – a very small group that committed the highest amount of crime. Though in this stage it is not clear how the further criminal behavior of this class will develop. In Moffit's taxonomy the life course persistent offender differs from the adolescent limited offender who ceases their offending in adulthood. Some delinquent behavior during adolescence can be considered normative and the majority of these phase out in the years to follow [20,53]. It is not possible to discern in this study which members of "High crime" class will cease their delinquency and who will retain it into adulthood. It could be that some of the members of this class will go on to develop a pattern of offending close to that of the life course persistent offender, but this is at this stage speculative.

Thirdly, although the classes of "High crime" and "Polydrug use and crime" displayed co-occurring behaviors, further solidifying the co-occurring nature of substance misuse and delinquency, their group sizes are small. This implies that when less frequent behaviors are not accounted for, i.e., occasional and one-time events of substance use or delinquency, only about 3% of the sample engaged in co-occurring substance use and delinquency. One interpretation of this finding can be that engagement in multiple risk behaviors during adolescence is perhaps less common than we think, when minor engagements are excluded.

Regarding associations with risk factors at T1. Risk factors were most salient for the "Alcohol only" class with 849 participants, where higher parental supervision and peer problems decreased the probability of belonging to this class, as compared to the "Low/abstainers" class. Lack of parental supervision has previously been established as a significant risk factor for problematic alcohol consumption among adolescents [23]. Considering peer problems, this measures social isolation and expulsion from friend groups of the same age. Peer influence is a strong risk factor for alcohol use and socializing with others who use alcohol increase the risk of consumption [17,18,23]. Thus, it is likely that less socializing with others decreased the probability. This does not necessarily mean that socializing with peers is something maladaptive. Although high alcohol consumption is associated with a number of health issues [1], not all adolescent drinkers retain their consumption into adulthood [55]. For some, temporary rule-breaking behavior along with peers can be considered a normative part of adolescence [20,55]. It seems that this pattern also can apply to alcohol use and after adolescence when peer influence is not as salient, the alcohol consumption will decrease among some members of the "Alcohol only" class. Hyperactivity, conduct problems and sensation seeking also increased the probability of belonging to the "Alcohol only" class, factors that have previously been associated with adolescent alcohol use [26]. Further, parental supervision and support decreased the probability of being classified into the "Polydrug use and crime" class whereas sensation seeking behavior and truancy increased the probability, factors that previously been associated with illicit drug abuse among adolescents [27,29]. In a similar matter, parental support decreased the probability of being classified into the "High crime" class whereas conduct problems increased the probability, where negative parental strategies and conduct problems, have previously been established to be related to delinquent behaviors [24,28]

When interpreting the results, consideration needs to be taken to the small group sizes of the "High crime" class and the "Polydrug use and crime" class that consisted of 32 and 82 individuals respectively, substantially limiting the statistical power. For example, the "High crime" class had considerable higher mean scores of sensation seeking behavior than the "Low/abstainers" class and the highest mean score of callous-unemotional traits, but neither of these risk factors remained significant, after the process of eliminating non-significant factors stepwise in the analytical approach. Thus, it is likely that factors that have previously been associated with criminal activity and substance use, such as callousness, impulsivity and sensation seeking behavior [25–29], did not reach significance in this study due to the lack of statistical power. Thus, these findings should be interpreted cautiously. Nevertheless, the risk factors that were significant in this study for the "High crime" and "Polydrug use and crime" classes, might be particularly salient due to the fact that they remained significant despite the apparent problem with lack of statistical power and low prevalence of drug use and delinquency in the sample.

Perhaps, the key to understand the "Polydrug use and crime" and "High crime" classes is similarities in risk profiles when it comes to severity. "High crime" and "Polydrug use and crime" members showed higher means and appears more maladjusted than "Alcohol only". For instance, the factors that predicted both "Alcohol only", "Polydrug use and crime" and/or "High crime" (parental supervision, sensation seeking behavior and conduct problems) had stronger association (weaker for parental supervision) to the latter groups. The latter groups also had higher levels of these risks than "Alcohol only". "Polydrug use and crime" and "High crime" also consistently had higher level of risk among all factors that reached significance for "Alcohol only", but not for the aforementioned classes. It has been emphasized previously that the most serious antisocial individual, such as the life course persistent offender in Moffit's taxonomy, display both problematic individual and familial factors [53]. Factors that both "Polydrug use and crime" and "High crime" displayed high levels of. Risk factors also have a cumulative effect, where more factors present leads to a higher risk of an outcome [56]. The "Polydrug use and crime" and "High crime" classes could possibly also have a wider range of different risk factors present, leading to problematic behaviors. In order to examine if risk factor severity and accumulation contributes to these more maladapted classes, a larger sample is required.

## Implications

There are some implications with these results. Firstly, since there are indications that extensive substance use and delinquency seems to be limited to small part of the general population and most are not in need of any interventional actions.

Interventions may be located and focused to where they are most needed instead. Though, this should be done with caution as targeted interventions may run the risk of labeling and stigmatizing individuals. Secondly, this study and research before indicate that the risk factor profile of the most severe substance users and delinquent offenders is multifactorial. The most extensive users and offenders have a high level and wide range of several different risk factors. It may be that interventions also need to be multifactorial and cover several different problem areas, such as improving mental health, family conditions, peer relations and school environment etc., in order to diminish and prevent further substance use and delinquent behaviors in the future.

### Strengths and Limitations

There are a number of strengths and limitations with this study. The major strength is the use of a randomly selected and nationally representative sample.

There are a number of limitations. The first one is the apparent lack of statistical power due to classes that displayed any extensive substance use and delinquency being very small, which might have had effects on the results, increasing type II errors. It could also be that participants that display very high degrees of substance misuse and delinquency could not be reached due to being incarcerated in juvenile correctional facilities, or rehabilitation centers, and thus were not included in this study. Hence, the results of this study might be biased towards those who do not engage in substance use or delinquency. Due to the low prevalence and how "other drugs" was measured, a further limitation was that it was not possible to make any comparison between different kind of substances (e.g., comparing sedatives to stimulants etc.) and how they co-occur and relate with delinquency. Another limitation refers to the use of self-report which relies on the recall of participants. There is also the risk of both under- and overreporting of crimes committed and substances consumed which might affect the reliability. Another limitation is that a relatively short time window has been investigated. The data only stretches for roughly two years between the ages of 15/16–17/18. It is possible that some risk factors may have been present in earlier stages of life which might be relevant from a developmental perspective, but also that latent class characteristics were present before risk factors. Also, there is no information beyond the age of 17/18, which may be relevant for understanding of further development.

With the inability to detect significant risk factors due to low prevalence of substance use and delinquency in the sample and limited statistical power, future studies should consider larger samples to reduce this problem. Another strategy for future research is to employ official registers such as medical records and crime registries. Since this study was biased towards those who did not engage in substance use and/or delinquency, official records could be possible way for studies with similar design to reach participants who could not be reached otherwise. Future studies might also want to consider comparing different types of substances and how they relate to and co-occur with delinquency. Also, longer time windows should be investigated which enables to examines developmental trajectories and differences between those whose substance use and delinquency is limited to adolescent years and those persist into later years.

### Conclusion

This study sought to identify different latent classes of substance misuse and delinquency in adolescence, and developmental risk factors associated with them. The vast majority of the participants do not engage in any extensive substance misuse or delinquency. The most common form of substance use was alcohol, likely adolescent binge drinking, and the most extensive drug use and delinquency were attributed about 3% of the sample. Substance users and delinquent offenders also show a multifactorial risk profile where several factors, such as familial problems, mental health issues, personality traits, peer factors and other factors are related to belonging to a class with substance use and/or committing delinquency. This has implications that interventions might need to cover several factors as well. A relatively short time window has been investigated and future research should investigate longer time frames to establish prevalence of which

limit their consumption/delinquency to adolescence and which persist into adult years. Statistical power was also a problem in this study and future research should use larger samples or other methods in an attempt to improve this.

## Supporting information

**S1 Table. Correlation matrix of indicator variables.**
(DOCX)

## Acknowledgments

The authors of this study have no specific acknowledgements to make.

## Author contributions

**Conceptualization:** Clas Björklund, Fredrik Sivertsson, Jonas Landberg, Peter Larm.

**Data curation:** Jonas Raninen.

**Formal analysis:** Clas Björklund.

**Investigation:** Jonas Raninen.

**Methodology:** Clas Björklund, Peter Larm.

**Supervision:** Peter Larm.

**Writing – original draft:** Clas Björklund.

**Writing – review & editing:** Fredrik Sivertsson, Jonas Landberg, Peter Larm.

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
