## [Decision Letter · Decision Letter 0]

18 Dec 2024

PONE-D-24-39038Latent classes of substance use and criminal offending in a Swedish national sample of adolescents and associated risk factorsPLOS ONE

Dear Dr. Björklund, 

Thank you for submitting your manuscript to PLOS ONE. After careful consideration, we feel that it has merit but does not fully meet PLOS ONE’s publication criteria as it currently stands. Therefore, we invite you to submit a revised version of the manuscript that addresses the points raised during the review process.

We look forward to receiving your revised manuscript.

Kind regards,

Ayse Ulgen, PhD, MGM

Academic Editor

PLOS ONE

Journal Requirements:

4. In this instance it seems there may be acceptable restrictions in place that prevent the public sharing of your minimal data. However, in line with our goal of ensuring long-term data availability to all interested researchers, PLOS’ Data Policy states that authors cannot be the sole named individuals responsible for ensuring data access (http://journals.plos.org/plosone/s/data-availability#loc-acceptable-data-sharing-methods).

5. Please include a caption for figure 1. 

6. We notice that your supplementary table S1 are included in the manuscript file. Please remove them and upload them with the file type 'Supporting Information'. Please ensure that each Supporting Information file has a legend listed in the manuscript after the references list.

Reviewers' comments:

Reviewer's Responses to Questions

**Comments to the Author**

1. Is the manuscript technically sound, and do the data support the conclusions?

Reviewer #1: Partly

Reviewer #2: Partly

Reviewer #3: Partly

2. Has the statistical analysis been performed appropriately and rigorously? 

Reviewer #1: I Don't Know

Reviewer #2: Yes

Reviewer #3: No

3. Have the authors made all data underlying the findings in their manuscript fully available?

Reviewer #1: Yes

Reviewer #2: Yes

Reviewer #3: Yes

4. Is the manuscript presented in an intelligible fashion and written in standard English?

Reviewer #1: Yes

Reviewer #2: Yes

Reviewer #3: Yes

5. Review Comments to the Author

Reviewer #1: Overall comments. Big sample, suitable methodological choice for the aim, good to focus on SU and criminal offending in synergy, and an important research question (including current relevance). However, there are some issues regarding lack of information on the procedure, definition of groups and interpretation of results that need to be resolved before being able to decide on the study’s contribution to the field (i.e. above all information regarding methodology, see comments below). The language is ok (with minor exceptions).

See attached file for specific recommendations.

Reviewer #2: The current study has potential but suffers from some important problems, which fortunately can be rectified.

1. The engagement of the literature is weak which is problematic because there are latent class papers of the crime/substance use overlap. Several of these are listed below and were based on the NESARC, NSDUH, and other epidemiological data.

a. Vaughn, M. G., Salas-Wright, C. P., Delisi, M., & Piquero, A. R. (2014). Health associations of drug-involved and criminal-justice-involved adults in the United States. Criminal Justice and Behavior, 41(3), 318-336.

b. DeLisi, M., Vaughn, M. G., Salas-Wright, C. P., & Jennings, W. G. (2015). Drugged and dangerous: Prevalence and variants of substance use comorbidity among seriously violent offenders in the United States. Journal of Drug Issues, 45(3), 232-248.

c. Fearn, N. E., Vaughn, M. G., Nelson, E. J., Salas-Wright, C. P., DeLisi, M., & Qian, Z. (2016). Trends and correlates of substance use disorders among probationers and parolees in the United States 2002–2014. Drug and alcohol dependence, 167, 128-139.

d. Kendler, K. S., Ohlsson, H., Sundquist, K., & Sundquist, J. (2013). A latent class analysis of drug abuse in a national Swedish sample. Psychological Medicine, 43(10), 2169-2178.

2. The discussion largely reiterates the findings and again the engagement of the literature is weak. It seems the findings have implications for Moffitt’s developmental taxonomy.

a. Moffitt, T. E. (1993). Adolescence-limited and life-course-persistent antisocial behavior: a developmental taxonomy. Psychological Review, 100, 674-701.

b. Moffitt, T. E. (2018). Male antisocial behaviour in adolescence and beyond. Nature human behaviour, 2(3), 177-186.

3. As the author notes, the cell sizes for polydrug use and crime and high crime latent classes are very small. Did you consider combining them? Perhaps forcing a 3-class solution makes more sense given that nearly 75% are abstainers and another 22% are normative alcohol only users.

Reviewer #3: A large nationwide study investigated the co-occurrence of self-reported substance use and criminal offending at ages 17/18, as well as whether a range of risk factors, including family, personality, mental health, peers, and others, measured at ages 15/16, predicted these co-occurrences. The findings revealed that it was possible to distinguish four classes of individuals based on their substance use (or non-use) and criminal behavior. Of these, only two classes were characterized by both substance use and criminal behavior, although they were very small. Specific factors appeared to be significant predictors of class membership. Overall, the topic is relevant to the field of study, although I have raised several points that should be addressed before this study can be considered for publication.

In general

Consider using the terms "delinquency" or "delinquent behavior" instead of "criminal behavior," as these terms are more appropriate when discussing adolescence.

Abstract

It would be useful to include the gender distribution in parentheses.

A large portion of the conclusion merely repeats the results. I recommend rewriting the conclusion section to focus on the implications of these findings, rather than restating the results.

Introduction

The first paragraph of the introduction could be strengthened by including more information on the global prevalence of both substance use and criminal behavior among adolescents, with a particular focus on Sweden, as this was the target population in your study. Including this background in the introduction can help better frame the outcomes of your study and underscore the relevance of your findings within both a global and national context.

In addition, within the same paragraph, when emphasizing the importance of your research, it would be beneficial to focus more specifically on the significance of studying the co-occurrence of substance use and delinquency in adolescence.

The continuation of the introduction could benefit from a more specific explanation of why adolescents are particularly susceptible to engaging in substance use and criminal behavior, along with a focus on the most common and important risk factors.

How were substance use and criminal behavior operationalized in your study?

In the section on the present study, it would be useful to specify which particular risk factors you focused on and provide a rationale for why these were selected. For example, if you chose to examine family dynamics, peer influence, or mental health problems, explaining how these factors are linked to both substance use and delinquency in your specific context would strengthen your research justification.

Methods

Participants and Procedure

Were there any inclusion or exclusion criteria for participation in this study?

Please provide more information on sample characteristics (e.g., mean age, ethnicity, SES, etc.) either in the text or in a table.

Measurements

The section on measures would benefit from additional subheadings to group measurements of the same category under a common heading.

Could you provide some examples of how you measured "Other drugs"?

You repeat the sentence "Risk factors were measured at T1" twice. Consider removing it from the first paragraph of the section on measures to avoid redundancy.

The reliability of the measure for callous-unemotional traits was poor. It would be useful to include the mean item correlation and indicate whether it falls within the acceptable range (for more detail, see Clark & Watson, 1995).

Clark, L. A., & Watson, D. (1995). Constructing validity: Basic issues in objective scale development. Psychological Assessment, 7(3), 309–319. https://doi.org/10.1037/1040-3590.7.3.309

Analysis

Please specify which software you used to conduct the latent class analysis.

Based on the literature, BIC is generally regarded as the most reliable criterion for determining the optimal number of classes. However, I recommend also considering entropy, classification error, class sizes, and whether the identified classes are meaningful and consistent with theoretical assumptions or prior research.

Correlations and other statistical findings should be reported in the Results section, not the Methods section.

I suggest that the reference group be mentioned in the Results section rather than in the Methods section, as at this point, it is not yet clear whether classes will be identified among the indicators.

Which software did you use to conduct the ANOVA analysis, and what was the initial number of risk factors considered?

Results

Please remove (AIC=52171.21, BIC=52377.40) from the text, as it is not informative on its own.

Table 1 should include additional relevant information, such as the number of parameters, likelihood ratio or log-likelihood, degrees of freedom, p-value, class error, and the entropy of R2.

In Table 2, you should also include the Wald test, as well as the Wald test for paired comparisons.

In the second paragraph of the Results section, the missing values in parentheses are surprising, given that it was previously mentioned that listwise deletion was employed.

In addition, to enhance readability, avoid placing all information in parentheses. You can write:

The largest class, which comprised 74.8 % of the sample (n = #, % females), scored the lowest on all outcomes and was therefore labeled the “Low/abstainers” class.

The third and fourth classes are very small, which raises questions about their meaningfulness. This relates to my earlier comment about providing more background information on the prevalence of substance use and delinquency in Sweden, as well as offering more detail in the Methods section regarding what class size would be considered acceptable.

Table 3 should include information on the F-test or Chi-Square test. Additionally, the Bonferroni correction should be applied for multiple pairwise comparisons.

The last paragraph of the Results section is unclear regarding which factors were excluded. To improve clarity and make it easier to follow, please mention the excluded factors in the text.

Discussion

In general, the discussion is somewhat limited and would benefit from a deeper analysis grounded in existing literature and empirical findings related to substance use and delinquency in adolescents.

Emphasizing that three classes were identified at T2 may cause confusion, as the first class was also derived from T2 measures. Clarifying this point would improve the paper's flow. Additionally, I recommend avoiding abbreviations and statistical terms in the discussion to enhance readability and clarity.

Alcohol is a significant risk factor for criminal behavior; however, in the "Alcohol only" class of your sample, it did not co-occur with criminal behavior. A broader discussion is needed to explore the possible reasons for this discrepancy. Is it due to the way criminal behavior was operationalized, or are there other factors that are stronger predictors of criminal behavior at this particular developmental stage than alcohol consumption alone? A more comprehensive exploration of these factors would be valuable.

In addition, when describing the third class, you could be more specific by stating that adolescents in this group were more likely to consume a variety of substances, with the highest probability of cannabis use. They were also more likely to commit crimes compared to both the reference category and the second class, although the difference was not so big. The results should be discussed in this context to provide a clearer understanding of the factors at play.

The discussion of the fourth class is superficial and should be grounded in evidence and prior research.

Considering the predictors of the classes, I suggest starting the discussion with the significant results, and then addressing the non-significant ones.

Instead of saying 'considerably higher OR,' explain what it means.

You should give more consideration to how measuring criminal behavior through self-report might influence your results.

Based on the limitations and your findings, are there any recommendations for future research?

What is the added value of your study, and what are the potential clinical implications?

Conclusion

The conclusion section should better summarize the main findings of the study and provide a clear take-home message.

6. PLOS authors have the option to publish the peer review history of their article (what does this mean? ). If published, this will include your full peer review and any attached files.

**Do you want your identity to be public for this peer review?** For information about this choice, including consent withdrawal, please see our Privacy Policy .

Reviewer #1: **Yes: ** Malin Hildebrand Karlén

Reviewer #2: No

Reviewer #3: **Yes: ** Marija Jankovic

---

## [Author Response · Author response to Decision Letter 0]

10 Feb 2025

Thank you for this opportunity for revision. Our responses to the reviewers suggestions can be found in the attached file "response to reviewers"

Thank you!

Sincerely

Clas Björklund

---

## [Decision Letter · Decision Letter 1]

24 Mar 2025

Latent classes of substance use and delinquency in a Swedish national sample of adolescents and associated risk factors

PONE-D-24-39038R1

Dear Dr. Bjorklund,

We’re pleased to inform you that your manuscript has been judged scientifically suitable for publication and will be formally accepted for publication once it meets all outstanding technical requirements.

Kind regards,

Ayse Ulgen, PhD, MGM

Academic Editor

PLOS ONE

Reviewers' comments:

Reviewer's Responses to Questions

**Comments to the Author**

1. If the authors have adequately addressed your comments raised in a previous round of review and you feel that this manuscript is now acceptable for publication, you may indicate that here to bypass the “Comments to the Author” section, enter your conflict of interest statement in the “Confidential to Editor” section, and submit your "Accept" recommendation.

Reviewer #2: All comments have been addressed

Reviewer #3: All comments have been addressed

2. Is the manuscript technically sound, and do the data support the conclusions?

Reviewer #2: Yes

Reviewer #3: Yes

3. Has the statistical analysis been performed appropriately and rigorously? 

Reviewer #2: Yes

Reviewer #3: Yes

4. Have the authors made all data underlying the findings in their manuscript fully available?

Reviewer #2: Yes

Reviewer #3: Yes

5. Is the manuscript presented in an intelligible fashion and written in standard English?

Reviewer #2: Yes

Reviewer #3: Yes

6. Review Comments to the Author

Reviewer #2: The author responded to reviewer comments thoroughly and thoughtfully. There are no lingering issues and the manuscript is significantly improved.

Reviewer #3: I do not have any further comments as all my concerns have been adequately addressed.

7. PLOS authors have the option to publish the peer review history of their article (what does this mean? ). If published, this will include your full peer review and any attached files.

**Do you want your identity to be public for this peer review?** For information about this choice, including consent withdrawal, please see our Privacy Policy .

Reviewer #2: No

Reviewer #3: **Yes: ** Marija Janković

---

## [Editor Report · Acceptance letter]

PONE-D-24-39038R1

PLOS ONE

Dear Dr. Björklund,

I'm pleased to inform you that your manuscript has been deemed suitable for publication in PLOS ONE. Congratulations! Your manuscript is now being handed over to our production team.

Kind regards,

on behalf of

Dr. Ayse Ulgen

Academic Editor

PLOS ONE